# Low-carbon optimal scheduling of integrated energy systems based on multi-strategy ameliorated goose algorithm and green certificate-carbon trading coordination

Chengcheng Ding, Yun Zhu *

School of Electrical Engineering, Guangxi University, Nanning, China

* 691524415@qq.com

## Abstract

The integrated energy systems (IES) in China face a dual challenge under the "dual-carbon" targets: maximizing renewable energy utilization while minimizing carbon emissions. Traditional tiered carbon markets often lack the flexibility to dynamically incentivize low-carbon operation. To address this, a coordinated framework is proposed, integrating a dynamic carbon emission trading (CET) mechanism with green certificate trading (GCT) and a Multi-Strategy Ameliorated Goose Optimization (MSAGOOSE) algorithm. The GCT-CET mechanism introduces exponential reward–penalty coefficients based on real-time renewable consumption rates, enabling adaptive carbon pricing. MSAGOOSE combines adaptive parameter adjustment, multimodal distribution-guided exploration, and population-aware reverse learning to improve optimization robustness in high-dimensional, nonlinear scheduling problems. Benchmark evaluations on CEC2017 and CEC2022 show that MSAGOOSE achieves an order-of-magnitude improvement in accuracy over seven state-of-the-art algorithms. In a 24-hour IES scheduling case in Anhui Province, the proposed method reduces carbon emissions by 27.3% (5,121 kg/d), increases renewable energy share to 88%, and cuts operating costs by 24.8% (6,151 CNY/d). Parametric analysis further confirms the framework's effectiveness in balancing economic and environmental goals under decentralized energy scenarios. This study presents a policy-algorithm co-design paradigm that offers both theoretical and practical support for low-carbon IES transitions, enabling scalable, flexible, and economically viable scheduling strategies.

## 1 Introduction

The global endeavor to achieve "dual-carbon" objectives poses intricate challenges for synchronized progress in governance structures, algorithm

**Data availability statement:** All relevant data are within the manuscript and its Supporting information files.

**Funding:** The author(s) received no specific funding for this work.

**Competing interests:** The authors have declared that no competing interests exist.

enhancement, and infrastructure administration to facilitate the global transition to cleaner energy [1]. China intends to to reach peak carbon emissions by 2030 and attain carbon neutrality by 2060, presenting a significant challenge to the transformation of the energy system. Studies [2] show that the integrated energy system (IES) is an effective carrier to achieve energy conservation and emission reduction, and low-carbon optimal scheduling as its core means can significantly improve energy efficiency and reduce carbon emissions. Improve energy efficiency, thereby promoting the green transformation of the energy system [3]. Therefore, this paper focuses on the low-carbon optimal scheduling of an integrated energy system.

Source-grid-load-storage Coordinated scheduling, using energy storage technology, synchronizes power and load resources, diminishes carbon footprints, increases new energy consumption capacity, and boosts system stability [4,5]. Reference [6,7], based on a non-Nash equilibrium, incorporates electricity-heat bidding within a distributed market-assisted recovery framework for multi-energy systems, providing a creative template for IES scheduling. Reference [8] developed a multi-time-scale optimal scheduling model to alleviate the effects of additional energy and load uncertainty on the system. However, the traditional IES scheduling is limited to a single economic or peak-valley difference target, which limits multi-objective coordination and system flexibility [9,10]. The conventional strategy inadequately accounts for environmental factors, particularly the emission of toxic substances ($SO_2$, $NO_x$, CO) [11–14]. It renders the current scheduling approach ill-suited for an "environmentally friendly society" and "double carbon" policy, as well as the significant transformations in the energy system [15,16]. This work presents an integrated energy model that accounts for the emission penalties of dangerous compounds, thereby mitigating their environmental harm.

China's carbon emission trading (CET) market, a crucial policy tool for emission reduction, exhibits improved cost-effectiveness through incremental carbon pricing compared to conventional carbon taxation [17,18]. Research [19] confirms the effectiveness of green certificate trading (CET) in increasing renewable energy integration. Progressive carbon pricing models improve the alignment of generation and demand in reducing emissions [20,21]. Current research on green certificate carbon trading is advanced yet critical, lacking a dynamic collaborative framework. The mechanism is fragmented, leading to increased costs of emission reduction as green electricity consumption rises [22,23]. In contrast, the static pricing mechanism fails to address the dual objectives of enhancing green electricity consumption and reducing carbon emissions [24,25]. The research suggests using asymmetric Nash bargaining to distribute benefits fairly in carbon market transactions, and its "contribution-bargaining ability" mapping technique may help integrate dynamic carbon markets [26]. This study formulates a dynamic GCT-CET coordination mechanism incorporating reward-penalty aspects associated with real-time renewable energy usage. The invention increases the sensitivity of emission-reduction incentives, leading to greater carbon reduction at elevated absorption levels and further promoting the achievement of the dual carbon goal.

In IES optimal scheduling, swarm intelligence algorithms are widely used, and their performance enhancement is crucial to meet the needs of energy development [27]. Genetic Algorithm (GA) and SOTA algorithms like GOOSE solve models well [28,29]. However, in high-dimensional scheduling, traditional swarm intelligence algorithms often suffer from premature convergence due to an imbalance between exploration and exploitation.The exponential growth of the search space, combined with fixed parameter settings, limits adaptability, leading to unstable convergence, reduced efficiency, and poor scalability [30–32]. The Improved Dung Beetle Optimization Algorithm (TDBO) employs adaptive parameter adjustment to achieve a balance between global and local optimization [33]. References [34,35] discuss the integration of global optimization (RJADE/TA) with a local search method, where the strategy is automatically adjusted based on the iterative proportion, leading to notable enhancements in convergence speed and result quality. This paper presents adaptive parameter adjustment for balancing global and local search, multimodal distribution guidance to avoid local optima, and population-aware reverse learning to improve high-dimensional exploration and algorithmic robustness, culminating in the formation of MSAGOOSE. This facilitates national carbon neutrality objectives by transforming policy instruments into practical scheduling solutions and enhancing algorithmic robustness for intricate energy-economic systems.

This paper's main contributions:

1. Developing an optimal scheduling model of integrated energy systems with toxic substance emission constraints.

2. Creating a dynamic green certificate – carbon trading linkage mechanism and conducting parameter sensitivity analysis to further reduce carbon emissions at high green electricity consumption levels and accelerate "dual-carbon" goals.

3. Applying MSAGOOSE, which uses adaptive parameter adjustment, multi-modal distribution guidance, and population-aware reverse learning for high-dimensional IES scheduling.

4. Based on parameter sensitivity and scheduling structure analysis, provide engineering suggestions and scheduling optimization recommendations.

Chapter 2 introduces the dynamic green certificate ladder carbon trading integrated energy model. Chapter 3 details the GOOSE algorithm and its enhancements. Chapter 4 compares the MSAGOOSE algorithm with other engineering algorithms utilizing the CEC2022 and CEC2017 datasets. Chapter 5 discusses the integration of the improved algorithm with an optimization model for case analysis. Chapter 6 concludes with a summary of research findings and a perspective on the future of the green certificate carbon trading market.

## 2 Dynamic green certificate-ladder carbon trading integrated energy system

### 2.1 Integrated energy model based on source-grid-load-storage

This paper focuses on an IES comprising wind power generation (WT), photovoltaic power generation (PV), diesel generator (DE), gas turbine (MT), battery energy storage system (BESS), fuel cell (FC), and grid interaction. The system structure is depicted in Fig 1, with detailed model information available in Reference [36]. The scheduling period is 24 hours (T = 24).

### 2.2 GCT – CET interaction model

CET is a mechanism that allows trading of carbon emission rights to control emissions through a legal identification system. GCT supports renewable energy portfolio standards. If a company's emissions exceed its quota, it must buy additional rights; otherwise, it can sell unused rights for profit [37]. The dynamic reward and punishment GCT-CET interaction mechanism is introduced in detail, and its structure is shown in Fig 2.

**2.2.1 Dynamic reward and punishment ladder carbon trading.** This research improves the conventional tiered carbon trading system by incorporating dynamic reward and penalty elements associated with levels of green power usage. This

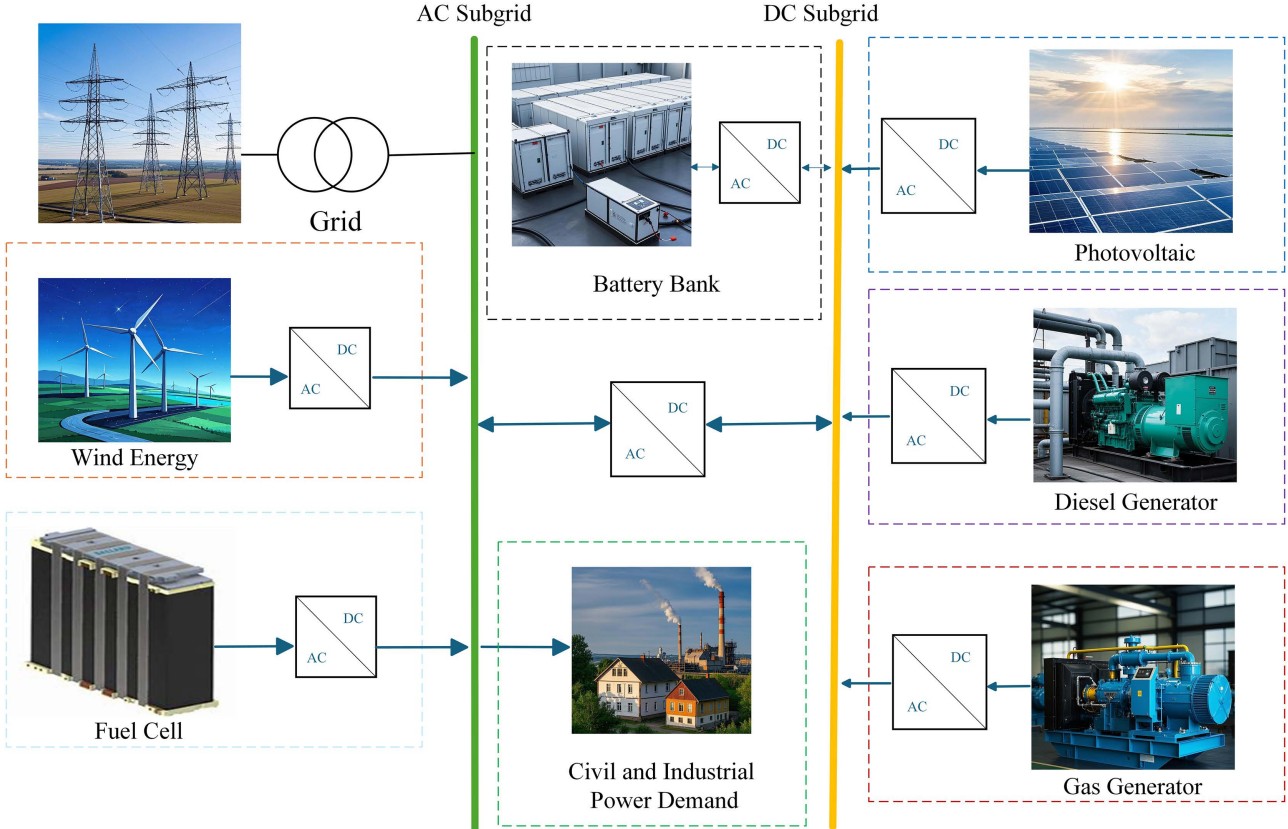

**Fig 1. Source-grid-load-storage IES operation framework.**

improvement intensifies the emphasis on carbon emissions and renewable energy usage in carbon credit trading, enabling the amalgamation of green certificates and carbon credit trading frameworks. The reform enhances the use of renewable energy, diminishes carbon emissions, and tackles the dual concerns of sustainable energy transition and environmental conservation.

$$
\begin{cases}
G_{IES} = G_{IES,a} - D_{IES}^{Total} \\
D_{IES}^{Total} = \sum_{t=1}^{24} \sum_{i=1}^{N} [d_i P_{i,t}] \\
G_{IES,a} = \sum_{t=1}^{24} \sum_{i=1}^{N} [a_i + b_i P_{i,t} + c_i P_{i,t}^2]
\end{cases}
\tag{1}
$$

In the formula, $G_{IES}$ represents the carbon emission trading amount, $G_{IES,a}$ is the actual carbon emission, $D_{IES}^{Total}$ is the carbon emission quota, $b_i$ is the carbon emission coefficient for each production equipment. Carbon emissions are primarily generated by Diesel Engines (DE), Grid interactions, Gas Turbines (WT), and Fuel Cells (FC). The specific parameters are shown in Reference [38].

In contrast to conventional tiered carbon trading pricing methods, dynamic reward-penalty elements amplify the impact of carbon emissions and green energy usage on trade results. As green electricity consumption rises and real carbon emissions exhibit a significant negative divergence from quotas (emissions falling below quotas), the incentive coefficient m escalates exponentially, leading to a corresponding increase in corporate earnings. Conversely, when green energy

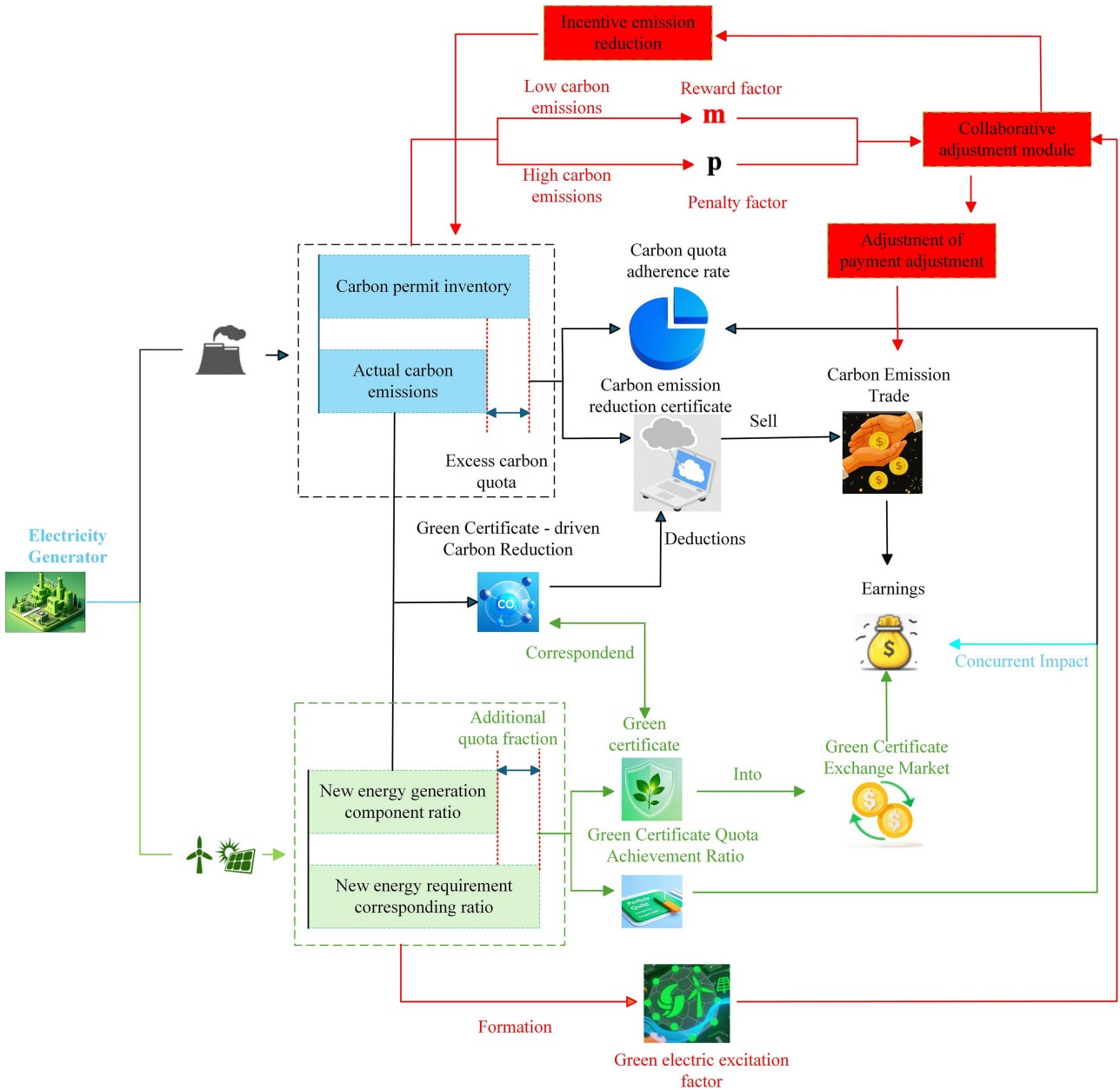

**Fig 2. Dynamic reward and punishment GCT-CET mechanism diagram.**

consumption is inadequate and carbon emissions exhibit a substantial positive deviation (emissions surpassing quotas), the penalty coefficient p escalates exponentially, markedly augmenting company expenses.

$$\begin{cases} \eta_{REE} = \left(\sum_{t=1}^{T=24}[P_{\mathrm{WT},t} + P_{\mathrm{PV},t}]\right)/\left(\sum_{t=1}^{T=24}[WT_t + PV_t]\right) & \eta_{REE} > 0.3 \\ m = \alpha(D_{IES}^{\mathrm{Total}}/G_{IES,a})^{\tan(\eta_{REE}\pi/3)} & G_{IES,a} \leq D_{IES}^{\mathrm{Total}} \\ p = \alpha(G_{IES,a}/D_{IES}^{\mathrm{Total}})^{\cot(\eta_{REE}\pi/3)} & D_{IES}^{\mathrm{Total}} \leq G_{IES,a} \end{cases} \quad (2)$$

In this formula, $m$ denotes the reward factor, $p$ denotes the penalty factor, and a denotes the reward-penalty coefficient. $\eta_{REE}$ represents the green power consumption rate, which is the ratio of the actual wind and solar energy supply to the wind and solar energy production capacity. $\eta_{REE}$ is generally greater than 30%. By using the tangent residual tangent function as an index, the power index is controlled within (0,4). This ensures the reward-penalty coefficient changes non-linearly with the green power consumption rate, aligning with emission reduction cost curves and enhancing economic incentives at higher consumption rates.

To further limit emissions, this paper sets up a segmented pricing mechanism. Step – by – step carbon trading divides emissions into intervals, with higher emissions incurring higher carbon emission rights prices and system costs, showing a step – by – step trend:

$$C_{CET} = \begin{cases} \lambda(1+m)(1+\mu)(G_{IES}+l)+\lambda l & -2l \le G_{IES} < -l \\ \lambda(1+m)G_{IES} & -l \le G_{IES} < 0 \\ \lambda(1+p)G_{IES} & 0 \le G_{IES} < l \\ \lambda(1+p)(1+\mu)(G_{IES}-l)+\lambda l & l \le G_{IES} < 2l \\ \lambda(1+p)(1+2\mu)(G_{IES}-2l)+\lambda(2+\mu)l & 2l \le G_{IES} < 3l \\ \lambda(1+p)(1+3\mu)(G_{IES}-3l)+\lambda(3+3\mu)l & 3l \le G_{IES} < 4l \\ \lambda(1+p)(1+4\mu)(G_{IES}-4l)+\lambda(4+6\mu)l & G_{IES} \ge 4l \end{cases} \tag{3}$$

In this formula, $C_{CET}$ is the step-wise carbon transaction cost, $\lambda$ is the base price of carbon trading, $l$ is the length of the carbon emission interval, and $\mu$ is the growth rate of the step-wise carbon transaction.

**2.2.2 Green certificate trading model.** Green Certificate Trading (GCT) uses green certificates to verify a company's renewable energy usage, helping implement renewable energy portfolio standards and market-based support instead of subsidies. GCT has two operations: enterprises buy certificates to avoid penalties if their renewable energy use is below the quota, and sell certificates for profit if usage exceeds the quota:

$$\begin{cases} C_{GCT} = \sigma_g \left( Q_i^s - Q_i^d \right) \\ Q_i^s = \alpha_g \sum_{t=1}^{T} (P_{LD}/1000) \\ Q_i^d = \kappa_g \sum_{t=1}^{T} [P_{WT,t} + P_{PV,t}]/1000 \end{cases} \tag{4}$$

Where $C_{GCT}$ is the green certificate trading cost; $Q_i^s$ is the required quota; $Q_i^d$ is the number of certificates from renewable energy; $\sigma_g$ is the trading price; $\alpha_g$ is the microgrid quota coefficient; $P_{LD}$ is the microgrid load at time t; $\kappa_g$ is the conversion coefficient (1 certificate = 1 MWh of new energy electricity); $P_{WT,t}$ and $P_{PV,t}$ are wind and solar outputs at time t.

**2.2.3 Green certificate-carbon joint trading mechanism.** The integration of GCT and CET mechanisms in this work is reflected in two key aspects: (1) Carbon emission reductions from renewable energy are accounted for in the carbon quota evaluation, where renewable generation offsets part of the emission allowance [38]. (2) Dynamic reward–penalty mechanisms based on green certificate attributes guide system scheduling toward increased renewable energy utilization. Green certificates participate in both CET and GCT systems, enabling their coordinated interaction and reinforcing low-carbon dispatch objectives. The precise stages of the joint mechanism are as follows:

1) According to Eq (1), the carbon emissions and carbon emission quotas of IES units are calculated.

2) According to the new energy data, the carbon emission reduction caused by the green certificate is calculated according to Eq (5), that is:

$$E_{green} = D_{carbon} - D_{green} \tag{5}$$

In the formula, $E$green represents the carbon emission reduction caused by new energy power generation; $D$green and $D$coal denote the carbon emission equivalents of new energy supply and carbon-emitting supply, respectively, over their respective periods.

3) According to the Eqs (1, 2), calculate the new carbon emission trading amount after the new energy offset and the coefficient of rewards and punishment.

$$G_{IES,\text{new}} = \left(G_{IES,a} - D_{IES}^{\text{Total}} - E_{\text{green}}\right) \tag{6}$$

### 2.3 Objective function

**2.3.1 Cost function.** This paper optimizes multi-regional IES using hybrid nonlinear programming. The objective function is delineated in Eqs (5–10) underscores the interaction between renewable energy deployment and the green certificate-carbon trading system. The function minimizes microgrid dispatching costs, including total maintenance, GCT-CET, environmental protection, and wind-solar abandonment penalties. The objective function is delineated as follows:

$$F_{total} = F_1 + F_2 + F_3 + F_4 \tag{7}$$

1) System maintenance cost

$$F_1 = \sum_{t=1}^{T} \sum_{i=1}^{N} [K_{\text{OM}} P_{i,t}] \tag{8}$$

Where $K_{\text{OM}}$ represents the maintenance coefficient for each micro - power source, and $P_{i,t}$ denotes the operating power of each power source, which includes WT, PV, MT, DE, BESS, FC, and energy interaction with the grid.

2) GCT-CET cost

$$F_2 = C_{CET} + C_{GCT} \tag{9}$$

Where $C_{CET}$ and $C_{GCT}$ represent the carbon emission trading cost and the green certificate trading cost, respectively.

3) Environmental protective cost

$$F_3 = \sum_{t=1}^{T} \sum_{i=1}^{N} \left(\mu_i \gamma_i P_{i,t}\right) \tag{10}$$

Where $\mu_i$ is the pollutant discharge coefficient, $\gamma_i$ is the cost of pollutant, and $P_{i,t}$ is the operating power of each power source. The pollutants studied are CO, $SO_2$, and $NO_x$.

**2.3.2 Constraint condition.** Stabilizing the IES requires accounting for power conservation constraints, power output constraints, Grid exchange constraints, and energy storage operation constraints, as described by the following equations:

(1) Power conservation constraints

$$P_{pv,t} + P_{wt,t} + P_{grid,t} + P_{DE,t} + P_{MT,t} + P_{bess,t} + P_{FC,t} = P_{L,t} \tag{11}$$

(2) Power output constraint

$$\begin{cases} P_{i,t}^{\min} \leq P_{i,t} \leq P_{i,t}^{\max} \\ |P_{i,t} - P_{i,t-1}| \leq R_i \Delta t \end{cases} \tag{12}$$

In the formula, $P_{i,t}^{\min}$ and $P_i^{\max}$ represent the upper and lower limits of each power output.

$R_i$ represents the limits of each power output.

(3) Energy storage operation constraint

$$\begin{cases} P_{bess}^{\min} \leq P_{bess,t} \leq P_{bess}^{\max} \\ Soc^{\min} \leq Soc_t \leq Soc^{\max} \end{cases} \tag{13}$$

In the formula, $P_{bess}^{\min}$ and $P_{bess}^{\max}$ are the minimum and maximum output power of the energy storage unit, respectively; $Soc^{\min}$ and $Soc^{\max}$ are the lower and upper limits of the capacity of the energy storage unit, respectively.

## 3 Multi-strategy ameliorated goose optimization algorithm

### 3.1 *Goose optimization algorithm*

Following the development of a dynamic green certificate-ladder carbon trading collaborative integrated energy model, an effective solution method is required for the model. For this task, this work enhances the goose optimization algorithm by a multi-strategy approach and creates the MSAGOOSE algorithm. This article will explain the MSAGOOSE algorithm's improvement method and benefits, and how it solves integrated energy system optimization models. The specifics of the original algorithm are presented in the reference [29].

**3.1.1 *Exploration Stage*.** In the exploration stage, the method uses random changes to make the search agent look at a lot of different solutions. It's important to calculate parameters (e.g., minimal time) and guide the search toward an optimal answer. Allowing the goose to randomly explore other agents' search space placements through random number generation is crucial. The variable sum is a key part of making the goose algorithm better at searching.

$$\begin{cases} X_{it,new} = randn(1, dim) * (M\_T * alpha) + Best\_X \\ alpha = 2 * (1 - (loop/MaxIt)) \end{cases} \tag{14}$$

Where *dim* represents the problem dimension, and *Best_pos* denotes the best position found in the search space so far, *MaxIt* denotes the maximum iterations allowed.

**3.1.2 *Development phase*.** During the utilization stage, the algorithm primarily conducts a fine – grained search within the neighborhood of the current optimal solution to enhance the solution's quality. This stage encompasses two key formulas. When $S\_W_{it} > 12$, the formula is as follows:

$$X_{it,new} = (T\_o\_A\_O_{it} * \sqrt{S\_W_{it}}/9.81) + D\_G_{it} * T\_A^2 \tag{15}$$

When $S\_W_{it} < 12$, the formula is as follows:

$$X_{it,new} = (T\_o\_A\_O_{it} * S\_W_{it}/9.81) * D\_G_{it} * T\_A^2 * c \tag{16}$$

In the above formula, $T\_o\_A\_O_{it}$ represents the object's arrival time, $D\_G_{it}$ denotes the distance of the goose, $T\_A$ is the average time, and $c$ is a coefficient used to adjust the search intensity.

The convergence condition of the algorithm is:

$$\begin{cases} \delta(k) = \frac{|Best^{(k)} - Best^{(k-1)}|}{Best^{(k)}}, \ \delta < \epsilon \ (\epsilon = 10^{-5}) \\ loop < MaxIt \end{cases} \tag{17}$$

When the relative change rate of the target value $\delta < \epsilon$ converges or the number of cycles is greater than *MaxIt* the cycle ends and is judged to be convergent. This is one of the most commonly used criteria for iterative algorithms.

### 3.2 *Goose algorithm improvement strategy*

This section proposes three coordinated strategies to address the key limitations of the GOOSE algorithm. Adaptive parameter adjustment (Section 3.2.1) guides global exploration, multimodal search (3.2.2) enhances diversity to avoid premature convergence, and reverse learning (3.2.3) improves local exploitation and convergence accuracy. Together, they form a logical closed loop of "parameter regulation → global exploration → local exploitation", effectively balancing exploration and exploitation in high-dimensional search spaces.

**3.2.1 *Adaptive adjustment factor.*** The GOOSE algorithm's conditional judgment parameters are fixed and cannot be modified to accommodate diverse challenges, thereby diminishing its efficiency. The adaptive adjustment strategy is integral to the whole iterative process, focusing on balancing global exploration with local development, optimizing the optimization path, and addressing convergence instability resulting from fixed parameters. The precise formula is as follows:

$$\begin{cases} p = \left| \frac{1-it}{\sqrt{1+it^2}} \right| + \frac{rand}{\sqrt{it}} \\ U = \left(1 - \frac{it}{MaxIt}\right) \cos\left(it \cdot \arccos(U)\right) \\ d = \left(d - \frac{d}{it \cdot MaxIt}\right) * U \end{cases} \tag{18}$$

The adaptive parameter $p$ decreases with iterations, switching to local search when $rnd \geq p$ and global search when $rnd < p$. This focuses on global exploration early and local exploitation later. Parameters $U$ and $d$ control the search perturbation, with $d$ reducing as iterations progress to refine the search towards the optimal solution.

**3.2.2 *Multimodal distribution guided walk strategy.*** The original GOOSE algorithm relies on a single random distribution during initial exploration, making it prone to local optima and imbalanced exploration–exploitation in high-dimensional, complex search spaces. This work introduces a Multimodal Distribution-Guided Random Walk (MDGRW) to overcome these limitations, enhancing global search robustness. MDGRW effectively balances locality and randomness, providing a generalized, extensible, and interpretable exploration framework. The subsequent sections will elucidate its principles, technical specifications, and application outcomes:

$$\begin{cases} X_{it,new} = Best\_pos + \Delta S * (M\_T * alpha), \quad \Delta S \sim \mathcal{P}_{selected}(it) \\ \mathcal{P}_{selected}(it) = \begin{cases} Gaussian(\mu, \sigma_t) & p \leq 1/3, LocalRenfinedDevelopment \\ Cauchy(x_0, \gamma) & p \geq 2/3, Mid-rangeLeap \\ Pareto(x_m, \alpha) & 1/3 < p \leq 2/3, GlobalWide-range \end{cases} \end{cases} \tag{19}$$

In the above formula, probability density functions (PDFs) define diverse variation modes, as Table 1, while a probability function $p$ dynamically selects strategies across search stages. As $p$ increases from 0 to 1, it prioritizes exploration early and exploitation later. The Gaussian distribution's superior local search capability enables fine-grained exploration of the search space, assisting the algorithm in discovering superior local solutions. The Cauchy distribution's heavy-tailed property facilitates large-scale jumps during global search, enabling effective escape from local optima. The Pareto distribution

**Table 1. The introduction of PDFs.**

| Distribution Type | Sampling Formula | Dynamic Parameter Adjustment | Role |
|---|---|---|---|
| Gaussian Distribution | $\Delta S \sim N(0, \sigma t2)$ | $\sigma t = \sigma 0 * e - 0.01 it$ | Fine-step convergence, suppresses oscillation |
| Cauchy Distribution | $\Delta S = \gamma * \tan(\pi(U - 0.5))$ | $\gamma = 2 - (it/T)$ | Heavy-tailed properties, escape local optima |
| Pareto Distribution | $\Delta s = Xm * (U - 1/\alpha - 1)$ | $\alpha = 1.5 + 0.5 \sin(\pi t/2\pi)$ | Long-distance jumps, explore unknown regions |

supports long-distance jumps, aiding the algorithm in exploring uncharted regions. By synergizing these three distributions, the algorithm balances global exploration and local exploitation, significantly enhancing its capability to locate global optimal solutions for complex optimization problems.

**3.2.3 *Population distribution-aware reverse learning strategy*.** In later iterations, the traditional GOOSE algorithm suffers from degraded computational efficiency in high-dimensional IES scheduling due to boundary stagnation. Search efficiency declines exponentially with dimensionality, causing performance degradation, exploration–exploitation imbalance, and entrapment in local optima. To address this, the proposed strategy dynamically adjusts search intensity using the population mean and an attenuation factor *d*, effectively mitigating efficiency loss and controlling computational complexity as the boundary expands with dimensionality. The algorithm's global optimization, resilience, and computational efficiency for high-dimensional problems improve significantly.

$$\begin{cases} X_{it,new} = X_{it} + \beta \otimes (Best\_X - d \cdot A) \\ A = \frac{1}{N} \sum_{i=1}^{N} X_{it,i} \end{cases} \tag{20}$$

In the above formula, *A* denotes the population mean. Mirror points dynamically adapt based on the current solution distribution. This adaptation mitigates the loss of search efficiency near problem boundaries. The term β introduces a stochastic perturbation. During initial iterations ($d \approx 1$), a larger deviation between the *Best_X* and the mean *A* indicates greater population diversity. This state promotes broader exploration of the search space. Conversely, in later iterations ($d \approx 0$), the population converges towards *Best_X*. This state intensifies localized search around the current best solution.

This paper addresses the deficiencies of the GOOSE algorithm by proposing three enhancement strategies aimed at improving its performance: the introduction of an adaptive adjustment factor for global regulation, a dynamic adjustment strategy for selecting parameter p (which governs the exploration-development strategy and MDGRW selection), and the optimization of the search range parameter d (which refines neighborhood exploration intensity). These strategies create a closed loop of ' parameter regulation→distribution exploration→neighborhood refinement ', effectively addressing the imbalance in high-dimensional space exploration and development, thereby facilitating the real-time and efficient resolution of IES scheduling cases. The flow chart of the MSAGOOSE algorithm is shown in Fig 3 and Algorithm 1.

**Algorithm 1. Multi-Strategy Ameliorated Goose Optimization**
```
Algorithm MSAGOOSE
Input: nPop, MaxIt, VarMin, VarMax, nVar, CostFunction
Output: Best_X, Best_Cost, Conv_curve
1: Initialize population using cubic chaotic map and reverse refraction
2: Evaluate initial fitness and find BestSol
3: for it=1 to MaxIt do
4:  Update adaptive parameters P, U, d
5:  Compute population mean A
6:  for each agent i do
7:   Apply Escape Strategy (if rand<P)
8:   Apply Multimodal Distribution Guided Walk Strategy:
9:    Generate S_W, T_o_A_O, T_o_A_S, T_T, T_A
```

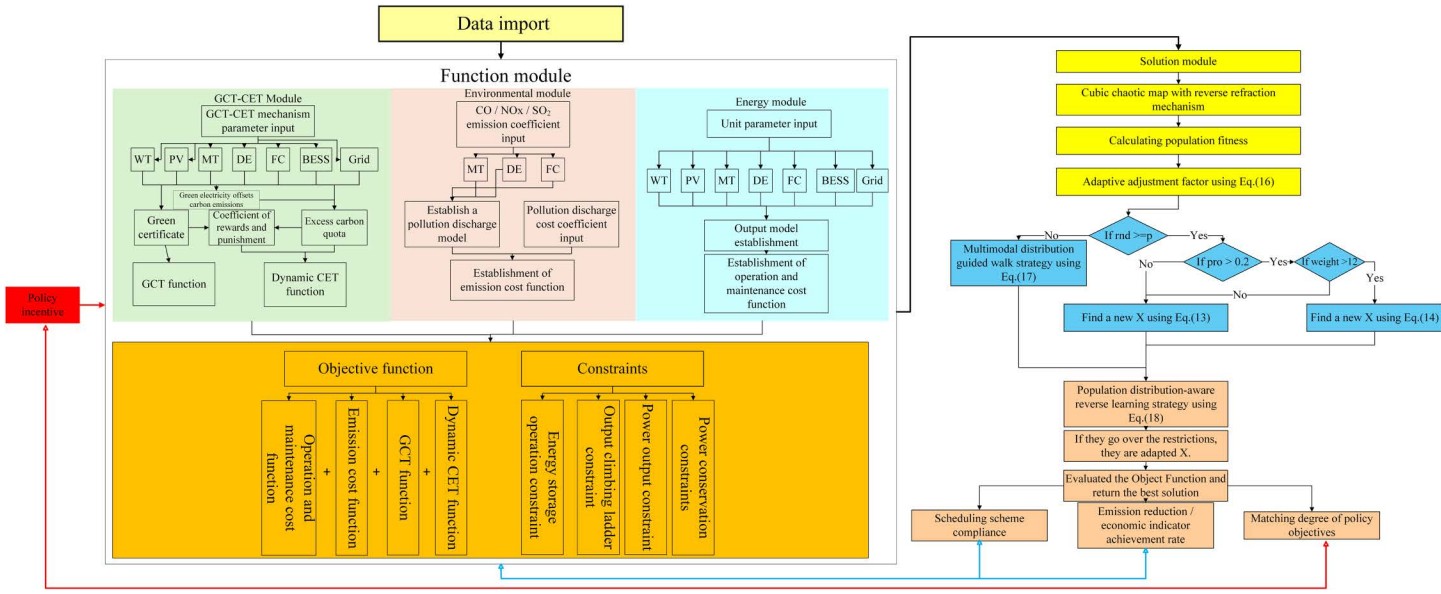

**Fig 3. The flow chart of the IES program solved by the MSAGOOSE algorithm.**

```
10:   if Exploration then
11:    Use PDFs for global search
12:   else if Exploitation then
13:    Use physical models (free fall, sound propagation)
14:   end if
15:   Apply Population Distribution-Aware Reverse Learning:
16:   if rand<P then
17:    X_{it,new}←Reflect(X_{it}, A,d)
18:   end if
19:   Clamp(X, VarMin, VarMax)
20:   Evaluate newsol.Cost
21:   if newsol.Cost<X_{it}.Cost then
22:    X←X_{it,new}
23:    Update BestSol if needed
24:   end if
25:  end for
26:  Record Conv_curve(it) = BestSol.Cost
27: end for
28: return Best_X, Best_Cost, Conv_curve
```

## 4 Algorithm performance test

The IES scheduling model features high dimensionality, nonlinear constraints (e.g., tiered carbon pricing, green certificate trading), and tightly coupled source–grid–load–storage interactions. These complexities hinder clear attribution of algorithmic performance in direct applications. To decouple algorithm behavior from model intricacies, the proposed method is first evaluated on the CEC2017 and CEC2022 benchmark suites, which reflect similar structural characteristics [39,40]. This pre-validation offers strong quantitative evidence of the algorithm's robustness and suitability for the real-world scheduling problem discussed in Section 5.

Experiments on CEC2022 and CEC2017 functions were conducted on a 3.70 GHz Intel Core i5-12600KF processor, 32 GB RAM, 64-bit Windows 10, and MATLAB R2024b system to confirm MSAGOOSE's performance and stability. For

a more accurate algorithm evaluation, CEC2017/2022's six most complex functions (F4/F8/F12/F7/F19/F30) encompass high-dimensional discontinuous, mixed, and composite types. The search space was established at 15,000 iterations and an initial population of 300. The results are stable and excellent, which verifies its reliability and portability in different scenarios. Each algorithm was executed 50 times, and essential metrics were documented. MSAGOOSE was compared to state-of-the-art algorithms, including GA, TDBO (2024), GSWOA, LEA, CPO, HO, and GOOSE, as well as ablation experiments with the multimodal distribution guided walk strategy (S1) and the population distribution-aware reverse learning strategy (S2), respectively.The convergence criterion is shown in the Eq (17). The selection of hyper-parameters is guided by the literature and CEC test function, and screened by the grid experiment method to ensure its reliability and portability in different scenarios. Table 2 introduces the selection of hyper-parameters of various algorithms. Table 3 presents the specific details of these functions [41,42], and the experimental results are shown in Table 4.

The $p$ denotes the Wilcoxon rank-sum test, a non-parametric check; when $p < 0.05$, we reject the null hypothesis and deem the improvement statistically significant rather than random. All *p-values* are far below $1 \times 10^{-6}$, confirming MSAGOOSE's superiority is highly significant across repeated runs. With a higher average optimal value and accuracy, the MSAGOOSE algorithm optimizes functions better than other recent algorithms. For the F4 function, MSAGOOSE exhibits a markedly lower average value, signifying enhanced global search capability and stability. The F12 function has the smallest standard deviation, indicating more steady performance across several runs and the trustworthy production of high-quality solutions, even though other algorithms have optimal values close to MSAGOOSE. In the ablation experiment, S1 and S2 techniques boost MSAGOOSE's optimization performance without increasing optimization time.

Fig 4 illustrates the iteration trajectories and box plots of all algorithms, effectively showcasing their performance and resilience following 50 tests. The MSAGOOSE algorithm demonstrably surpasses other algorithms in terms of resilience and optimization efficacy. The implementation of the Population Distribution-Aware Reverse Learning Strategy improves MSAGOOSE's capacity to optimize high-dimensional problems in the CEC2017 (dim = 100) test functions, namely in F19 and F30.

Table 2. Hyperparameters of various algorithms.

| Algorithm | Hyperparameters | Convergence Criterion |
|---|---|---|
| GA | p_c = 0.9, p_m = 0.01 | Loops = *Maxit* or $\delta$ < 1e-6 over 30 iterations |
| TDBO | P = 0.2; k = 5; w1 = 0.5; w2 = 0.1 | Loops = *Maxit* or $\delta$ < 1e-6 over 30 iterations |
| GSWOA | w_max = 0.2–0.5; b_base = 0.75; P = 0.3–0.7 | Loops = *Maxit* or $\delta$ < 1e-6 over 30 iterations |
| LEA | h_max = 0.7; h_min = 0; λ_c = 0.5; λ_p = 0.5 | Loops = *Maxit* or $\delta$ < 1e-6 over 30 iterations |
| CPO | T = 2; step = 0.02; α = 0.2; Tf = 0.8 | Loops = *Maxit* or $\delta$ < 1e-6 over 30 iterations |
| HO | b∈ [2 4 ]; c∈[1, 1.5]; d∈ [2 3 ]; l∈[−2π, 2π] | Loops = *Maxit* or $\delta$ < 1e-6 over 30 iterations |
| GOOSE | coe = 0.83; Weight_Stone = 12; pro = 0.2; rnd = 0.5 | Loops = *Maxit* or $\delta$ < 1e-6 over 30 iterations |
| MSAGOOSE | coe = 0.83; Weight_Stone = 12; pro = 0.2; rnd = p | Loops = *Maxit* or $\delta$ < 1e-6 over 30 iterations |

Table 3. CEC2022 and CEC2017 benchmark functions.

| Category | No. | Functions | Fi* |
|---|---|---|---|
| CEC2022 (Dim = 20) | 4 | Shifted and full Rotated Non-Continuous Rastrigin's Function | 800 |
| | 8 | Hybrid Function 3 (N = 5) | 2200 |
| | 12 | Composition Function 4 (N = 6) | 2700 |
| CEC2017 (Dim = 100) | 7 | Shifted and Rotated Lunacek Bi-Rastrigin Function | 700 |
| | 19 | Hybrid Function 6 (N = 5) | 1900 |
| | 30 | Composition Function 10 (N = 3) | 3000 |
| | | Search range: [−100,100] | |

**Table 4. Experimental results on CEC2022 and CEC2017.**

| Functons | Algorithms | | | | | | | | | | |
|---|---|---|---|---|---|---|---|---|---|---|---|
| | | GA | S2 | TDBO | S1 | GSWOA | HO | CPO | LEA | GOOSE | MSAGOOSE |
| F4 | Opt | 898 | 841 | 899 | 856 | 907 | 839 | 916 | 855 | 891 | 800 |
| | Ave | 933 | 858 | 923 | 882 | 931 | 886 | 921 | 873 | 901 | 801 |
| | Std | 1.43 | 15.27 | 17.08 | 13.9 | 21.86 | 10.22 | 26.18 | 15.17 | 23.1 | 24.42 |
| | Fri | 8.7 | 2.75 | 7.9 | 4.15 | 8.8 | 4.35 | 7.9 | 3.45 | 5.95 | 1.05 |
| | Rank | 9 | 2 | 7 | 4 | 10 | 5 | 8 | 3 | 6 | 1 |
| | p | 2.33E-11 | 2.33E-11 | 2.33E-11 | 2.33E-11 | 2.33E-11 | 2.33E-11 | 2.33E-11 | 2.33E-11 | 2.33E-11 | |
| F8 | Opt | 2256 | 2223 | 2246 | 2227 | 2229 | 2239 | 2240 | 2249 | 2455 | 2200 |
| | Ave | 2299 | 2259 | 2251 | 2229 | 2238 | 2241 | 2241 | 2278 | 2531 | 2201 |
| | Std | 63.82 | 57.11 | 6.11 | 5.28 | 7.84 | 9.43 | 2.93 | 53.95 | 70.52 | 0.33 |
| | Fri | 7.05 | 4.10 | 7.30 | 2.85 | 5.00 | 5.20 | 5.95 | 6.55 | 10.00 | 1.00 |
| | Rank | 8 | 3 | 9 | 2 | 4 | 5 | 6 | 7 | 10 | 1 |
| | p | 2.96E-07 | 2.96E-07 | 2.96E-07 | 2.96E-07 | 2.96E-07 | 2.96E-07 | 2.96E-07 | 2.96E-07 | 2.96E-07 | |
| F12 | Opt | 2914 | 2801 | 2795 | 2836 | 2947 | 2837 | 3105 | 2892 | 4601 | 2700 |
| | Ave | 3158 | 2890 | 2884 | 2889 | 3533 | 2899 | 2962 | 2861 | 5468 | 2732 |
| | Std | 72.12 | 113.36 | 32.57 | 44.20 | 220.85 | 42.64 | 12.12 | 21.01 | 387.59 | 9.65 |
| | Fri | 8.00 | 3.50 | 4.20 | 4.30 | 9.00 | 4.60 | 6.55 | 3.15 | 10.00 | 1.70 |
| | Rank | 8 | 3 | 4 | 5 | 9 | 6 | 7 | 2 | 10 | 1 |
| | p | 3.02E-11 | 3.82E-09 | 3.02E-11 | 3.82E-09 | 3.02E-11 | 3.02E-11 | 3.02E-11 | 3.02E-11 | 3.02E-11 | |
| F7 | Opt | 2250 | 739 | 2460 | 1486 | 3012 | 2579 | 1778 | 2317 | 2392 | 703 |
| | Ave | 2436 | 799 | 2533 | 1613 | 3055 | 2632 | 1812 | 2431 | 2449 | 715 |
| | Std | 370.73 | 80.28 | 146.92 | 253.24 | 86.19 | 105.38 | 67.88 | 227.07 | 112.70 | 76.23 |
| | Fri | 6.57 | 1.87 | 7.10 | 3.23 | 9.90 | 8.13 | 3.77 | 6.73 | 6.57 | 1.17 |
| | Rank | 5 | 2 | 8 | 3 | 10 | 9 | 4 | 7 | 6 | 1 |
| | p | 3.02E-11 | 3.02E-11 | 3.02E-11 | 3.02E-11 | 3.02E-11 | 3.02E-11 | 3.02E-11 | 3.02E-11 | 3.02E-11 | |
| F19 | Opt | 2049 | 2009 | 2001 | 1979 | 1988 | 1991 | 1991 | 2028 | 2281 | 1951 |
| | Ave | 8.08E+06 | 3.85E+03 | 9.08E+08 | 1.82E+06 | 2.08E+10 | 1.63E+07 | 3.56E+07 | 1.28E+07 | 8.04E+05 | 2.00E+03 |
| | Std | 1.23E+06 | 2.43E+03 | 2.45E+07 | 1.69E+05 | 2.08E+08 | 9.94E+05 | 1.14E+06 | 6.01E+05 | 1.08E+05 | 1.04E+03 |
| | Fri | 5.33 | 1.83 | 9.00 | 3.93 | 10.00 | 6.43 | 7.77 | 6.27 | 3.27 | 1.17 |
| | Rank | 5 | 2 | 9 | 4 | 10 | 7 | 8 | 6 | 3 | 1 |
| | p | 3.02E-11 | 1.86E-10 | 3.02E-11 | 1.86E-10 | 7.69E-11 | 3.02E-11 | 3.02E-11 | 3.02E-11 | 3.02E-11 | |
| F30 | Opt | 6.38E+06 | 5.09E+03 | 8.60E+08 | 7.23E+05 | 1.69E+10 | 1.24E+08 | 1.96E+08 | 1.84E+07 | 2.02E+06 | 5.09E+03 |
| | Ave | 4.90E+07 | 1.01E+04 | 2.32E+09 | 2.09E+07 | 2.36E+10 | 2.69E+08 | 3.35E+08 | 7.23E+07 | 2.91E+06 | 9.98E+03 |
| | Std | 3.08E+07 | 5.86E+03 | 7.53E+08 | 2.35E+07 | 5.26E+09 | 9.75E+07 | 8.60E+07 | 2.96E+07 | 4.64E+05 | 5.43E+03 |
| | Fri | 5.10 | 1.90 | 9.00 | 4.13 | 10.00 | 7.23 | 7.77 | 5.63 | 3.13 | 1.10 |
| | Rank | 5 | 2 | 9 | 4 | 10 | 7 | 8 | 6 | 3 | 1 |
| | p | 1.88E-11 | 5.83E-10 | 2.56E-11 | 5.83E-10 | 1.88E-11 | 2.41E-11 | 2.51E-10 | 1.66E-11 | 1.88E-10 | |

The Multimodal Distribution-Guided Walk Strategy enhances the optimization performance of MSAGOOSE for F8, resulting in increased accuracy and stability. When the issue dimension increases from 20 to 100, MSAGOOSE's convergence accuracy reduces by 8.7%, while GOOSE's decreases by 63.2%, proving the better dimension disaster technique works.

The effectiveness and practicality of MSAGOOSE in addressing complex the IES optimization and scheduling problems are highlighted. The adoption of the proposed GCT-CET fusion strategy not only delivers excellent economic and environmental outcomes but also strongly supports the sustainable development of the system.

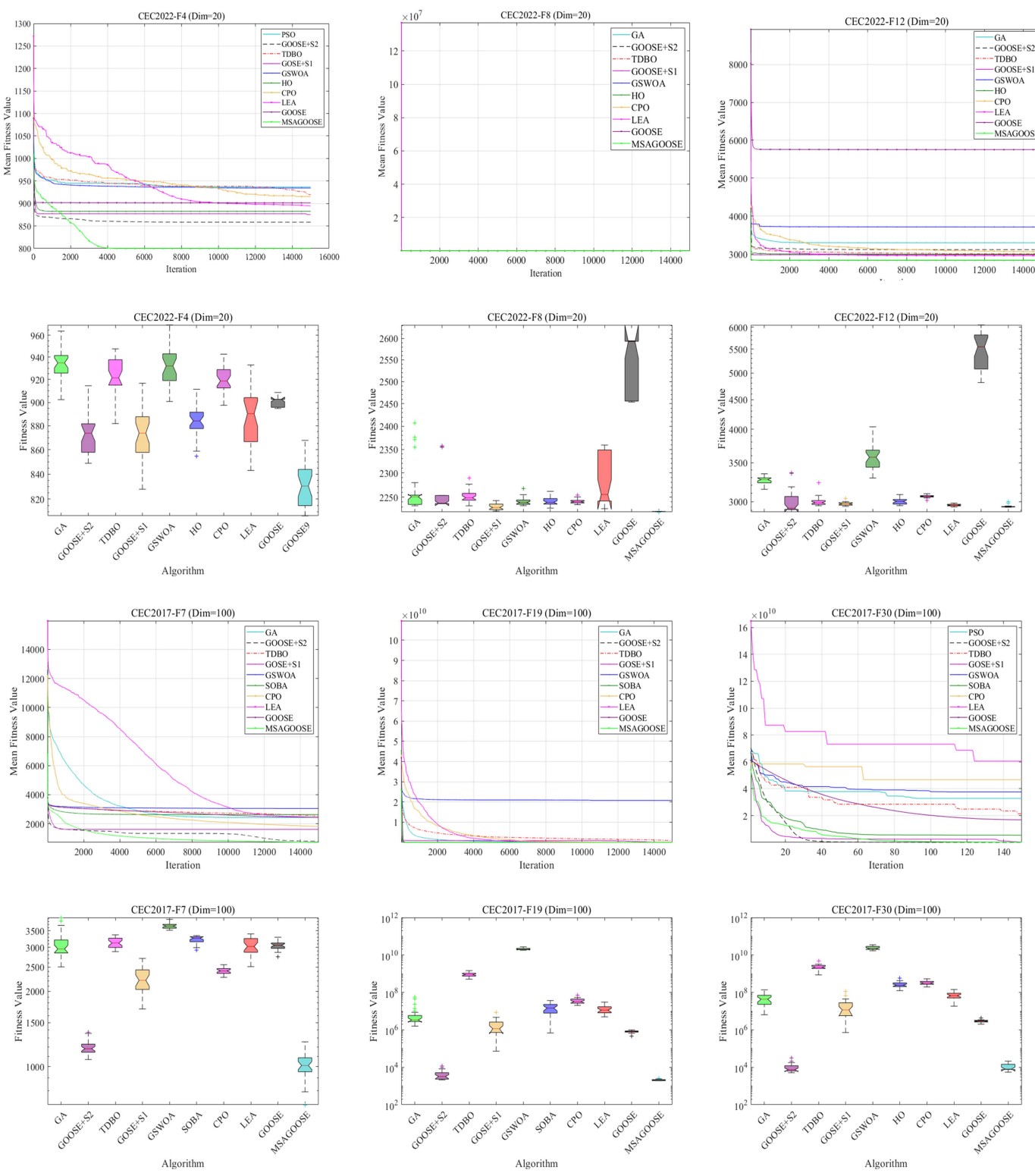

**Fig 4. Results of MSAGOOSE and other algorithms on CEC2022 and CEC2017.**

## 5 Case study analysis

This paper uses an operating RIES in Anhui Province, China, as a simulated case study. RIES covers wind, solar, gas, diesel, fuel cell, and energy storage systems. The scheduling cycle is 24 hours, and a seven-day, 24-hour, high-dimensional, difficult optimization scenario is developed. Historical data shows a 10% variation in wind and photovoltaic power output. Figs 3 and 4 illustrate the deviation ranges of solar and wind power generation alongside various loads. Fig 5 displays IES solar and wind power generation and load demand deviation ranges.

Table 5 lists time-of-use (TOU) electricity prices, with natural gas at 0.35 CNY/kWh, and stepped carbon trading parameters (base price: 0.25 CNY/kg, 25% incremental rate, 2,000 kg allowance intervals, $\alpha = 0.3$ =0.3). Table 2 outlines the operational parameters of energy supply equipment, while Table 1 details the daily stepped power pricing structure and presents the pollutant emission coefficients of generation units. The case configurations are detailed in Tables 6–9.

The MSAGOOSE algorithm was employed to solve this model. The maximum number of iterations was set to 3,000, and the initial population size was 300, to ensure an accurate computational result.

### 5.1 *Algorithm performance*

Further validation is needed to see if MSAGOOSE can be applied to real-world engineering models, despite its impressive optimization capabilities across benchmark functions. The green certificate-carbon trading (GCT-CET) model is used to evaluate its performance in Scenes 4 and 5 (S4/S5). The maximum iterations and population sizes of all algorithms were uniform for fairness. Comparative results are displayed in the Table 10 and Fig 6.

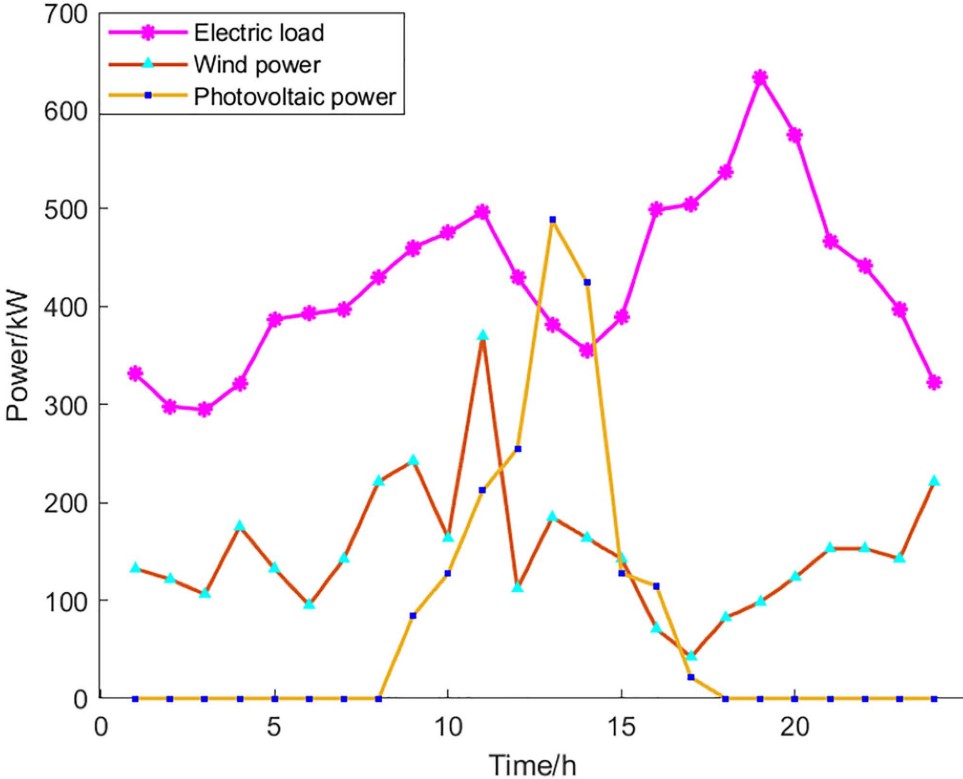

**Fig 5. Load forecasting curves of typical users.**

**Table 5. Time-of-use electricity price.**

| Period | Time | Purchase (CNY/kWh) | Sell (CNY/kWh) |
|---|---|---|---|
| Valley Period | 0:00-7:00;23:00-24:00 | 0.52 | 0.39 |
| Normal Period | 7:00-10:00;15:00-18:00;21:00-23:00 | 0.94 | 0.51 |
| Peak Period | 10:00-15:00;18:00-21:00 | 1.24 | 1.06 |

**Table 6. Pollutant treatment cost and emission coefficient [43].**

| Pollutant | Pollutant treatment costs | Discharge ratio(g/kWh) | | | |
|---|---|---|---|---|---|
| | | MT | FC | Grid | DE |
| CO | 10.17 | 0.049 | 0 | 0.082 | 435.2 |
| $NO_x$ | 13.213 | 0.29 | 0.019 | 1.47 | 10.09 |
| $SO_2$ | 67.138 | 0.0039 | 0.0035 | 1.34 | 0.306 |

**Table 7. Related operating parameters of power supply equipment.**

| Type | DE | MT | FC | Grid |
|---|---|---|---|---|
| Lower power limit/kW | 3 | 3 | 3 | −300 |
| Upper power limit/kW | 150 | 150 | 150 | 300 |
| Limit of climbing power/(kW/min) | 1.5 | 1.5 | 1.5 | |
| Operation coefficient (CNY/kW) | 0.128 | 0.0489 | 0.0288 | |

**Table 8. Energy storage parameters.**

| Type | Parameter | Value | Parameter | Value |
|---|---|---|---|---|
| BESS | $Soc_{max}$/ (kWh) | 150 | $P_{BESS, min}$/ kW | −150 |
| | $Soc_{min}$/ (kWh) | 0 | Efficiency | 0.9 |
| | $P_{BESS, max}$/ kW | 150 | | |

**Table 9. Scene setting information.**

| Scene | Operating Cost | Environment Penalty | CET | GCT | Improved CET Cost |
|---|---|---|---|---|---|
| Scene 1 | ✓ | | | | |
| Scene 2 | ✓ | ✓ | | | |
| Scene 3 | ✓ | ✓ | ✓ | | |
| Scene 4 | ✓ | ✓ | ✓ | ✓ | |
| Scene 5 | ✓ | ✓ | | ✓ | ✓ |

**Table 10. Comparison of the convergence curves between the algorithms.**

| | | GA | GOOSE +S2 | TDBO | GOOSE +S1 | GSWOA | HO | CPO | LEA | GOOSE | MSAGOOSE |
|---|---|---|---|---|---|---|---|---|---|---|---|
| Scene 4 | Opt/CNY | 2.05E+05 | 9.99E+03 | 5.47E+04 | 4.45E+04 | 9.89E+04 | 2.25E+04 | 3.09E+05 | 5.20E+05 | 8.05E+04 | 7.94E+03 |
| | time/s | 153.23 | 152.11 | 169.45 | 168.4 | 167.4 | 171.4 | 165.34 | 163.24 | 154.8 | 155.33 |
| Scene 5 | Opt/CNY | 2.42E+05 | 8.26E+03 | 4.91E+04 | 2.44E+04 | 8.46E+04 | 3.46E+04 | 3.43E+05 | 4.97E+05 | 7.86E+04 | 6.15E+03 |
| | time/s | 153.65 | 153.15 | 165.72 | 166.4 | 164.9 | 164.4 | 163.83 | 162.78 | 155 | 155.73 |

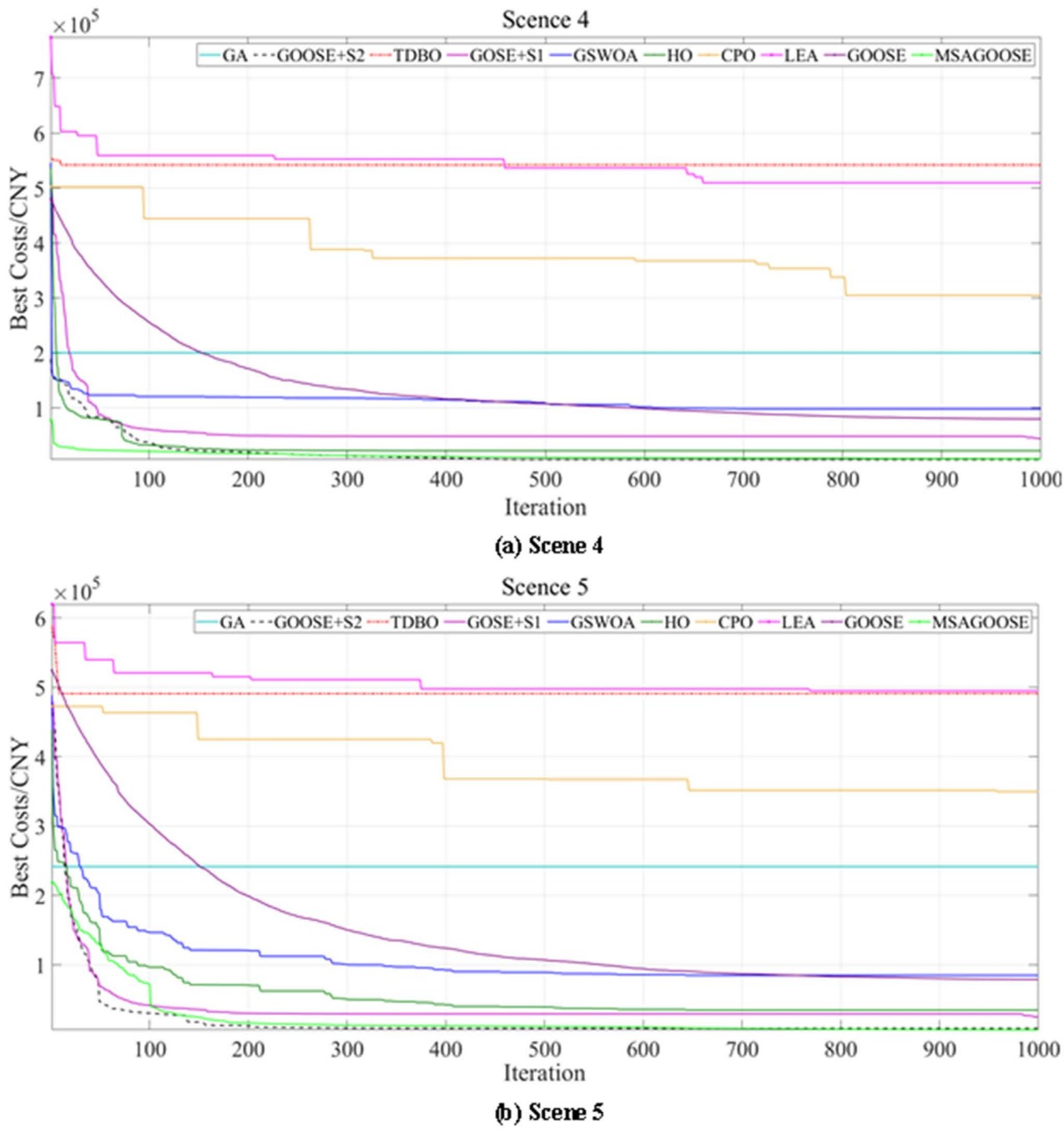

**Fig 6. Comparison of multi-algorithm convergence curves.**

When compared to nine other algorithms, MSAGOOSE performs better in both average and optimal values, thanks to its higher-quality starting population and optimization capabilities. It escapes local patterns at iteration 100 and converges later, demonstrating global exploration. Additionally, it is efficient enough to satisfy the demands of real project scheduling, as evidenced by its shorter computational durations of 155.33s for Scene 4 and 155.73s for Scene 5.

**Table 11. Comparison of multi-scene operation results.**

| Scene | 1 | 2 | 3 | 4 | 5 |
|---|---|---|---|---|---|
| Carbon emissions/kg | 18894.91 | 11634.36 | 5887.19 | 5840.89 | 5121.34 |
| SO2/kg | 3.44 | 2.92 | 3.77 | 3.35 | 2.60 |
| NOX/kg | 33.80 | 4.88 | 6.42 | 5.94 | 4.95 |
| CO/kg | 1298.60 | 35.43 | 64.32 | 63.52 | 32.82 |
| Green power rate | 0.79 | 0.77 | 0.74 | 0.83 | 0.88 |
| Operating cost/CNY | 8183.37 | 7892.06 | 7332.80 | 7294.49 | 6658.94 |
| Carbon trading cost/CNY | | | 341.31 | 466.94 | −270.67 |
| Green certificate cost/CNY | | | | −770.86 | −841.85 |
| Environment Penalty/CNY | | 620.90 | 1003.54 | 946.95 | 604.80 |
| Total cost/CNY | 8183.37 | 8512.96 | 8677.65 | 7937.51 | 6151.23 |

## 5.2 Analysis of multi-scene operation results

This study's scheduling model minimizes the integrated energy system's total cost, encompassing operation, environmental, curtailment, and green certificate-carbon trading costs. Simulation results across Scenes are in Table 11, with positives as costs and negatives as revenues.

Scene 1 and Scene 2: Scene 2 indicates a cost escalation of 329.59 CNY, attributed to a 620.90 CNY increase in environmental penalty expenses. $SO_2$ diminishes by 524.58 g, $NO_x$ diminishes by 29,059.61 g, and CO diminishes by 1,263,176.8 g. Pollutant emissions are reduced, indicating partial regulation of environmental effects despite increased expenses. The coal consumption scheduling mode, which prioritizes minimizing system coal usage, is implemented, sacrificing economic efficiency for enhanced environmental metrics.

Scene 2 and Scene 3: Lower operation expenses and an 8.7% decrease in wind/solar abandonment rate (increasing green power utilization) reduce costs by 164.69 CNY, compensating renewable energy curtailment penalties. Pollution rebound transpires, as elevated emissions result in rising penalty costs, highlighting inconsistencies in emission control methods or modifications to the energy system.

Scene 3 and Scene 4: The introduction of green certificates and carbon trading systems in Scene 4 results in a total cost reduction of 740.14 CNY, a carbon emissions reduction of 46.30 kg, and a green power ratio increase to 0.83. Environmental fines decreased to 946.95 CNY. GCT-CET fosters preferred scheduling of low-carbon units, over-quota renewable energy consumption, considerable green power consumption, emission reductions, and market-based techniques that cover environmental costs to generate economic benefits.

Scene 4 and Scene 5: Scene 5 showcases the most effective use of carbon trading and synergistic green certificates, leading to even greater reductions in carbon emissions, even at very high levels of green power usage. Scene 1 demonstrates elevated green power usage, yet does not significantly mitigate hazardous emissions and carbon output. Emissions of CO, $SO_2$, and $NO_x$ are cut by 75.4%, 24.3%, and 85.4%, respectively, thanks to the market mechanism and multi-energy complementarity working in tandem. Scene 3 prioritizes carbon emissions in the objective function, not increased green electricity usage. The twin systems provide logical energy allocation, maximizing both ecological and financial advantages.

## 5.3 Sensitivity analysis of improved carbon trading model

The results of parameter sensitivity analysis are shown in Fig 7.

The IES demonstrates varied reactions to diverse tiered carbon trading coefficients. Current research mostly examines the effects of carbon trading parameters on total system costs while neglecting the direct consequences on carbon

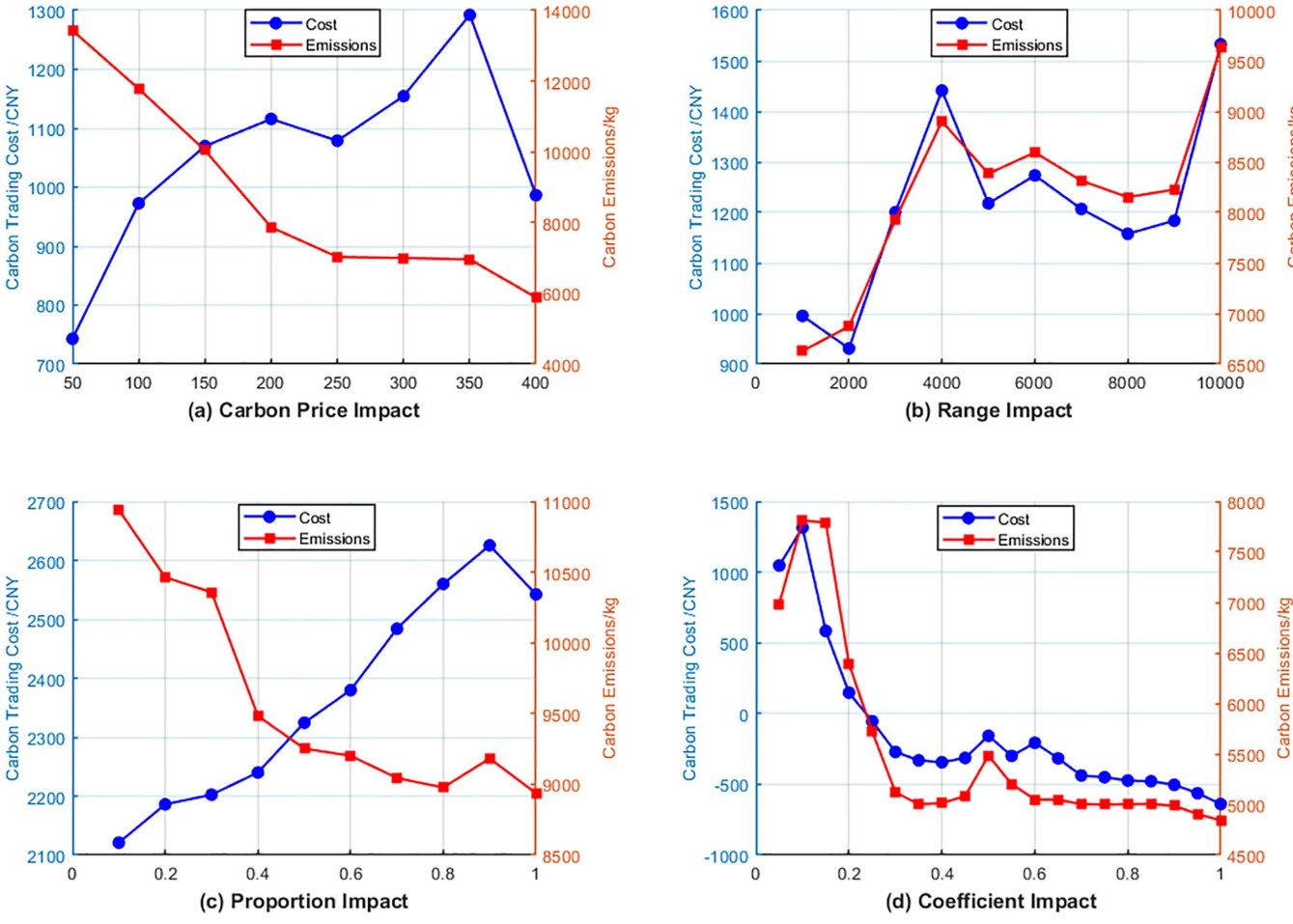

**Fig 7. Sensitivity analysis.**

emissions and trading expenses. Examining the impact on system carbon emissions and trading costs as well as the underlying mechanisms, this study examines how different benchmark carbon prices, price growth rates, interval settings, and reward-penalty coefficients affect IES performance.

As illustrated in Fig 7, the algorithm optimizes emission reduction scheduling when the benchmark price is below 200, since an increase reduces emissions and raises carbon prices. Above 400, equipment output adjustments limit emission decreases, forcing the system to forgo economic efficiency to meet carbon constraints. This leads to increased overall expenses with less effectiveness in carbon reduction.

Widening the price range increases volatility in carbon allowance prices. Expanding the price range increases the volatility of carbon allowance prices. Smaller ranges restrict high-carbon unit operation below 2000, whereas longer intervals ease constraints, reactivating thermal power and generating emission rebounds. The system's cost-balancing results in increased cost volatility and persistent emission growth.

The growth rate of carbon trading prices affects the extent of tiered rises in carbon trading costs within the model. The system slowly adjusts equipment output to reduce rising carbon trading costs when the growth proportion is below 0.3. Higher carbon allowance prices lead the system to cut emissions to control total costs at growth rates between 0.3 and 0.8, but equipment output restrictions limit considerable emission reductions, and carbon trading costs rise with price

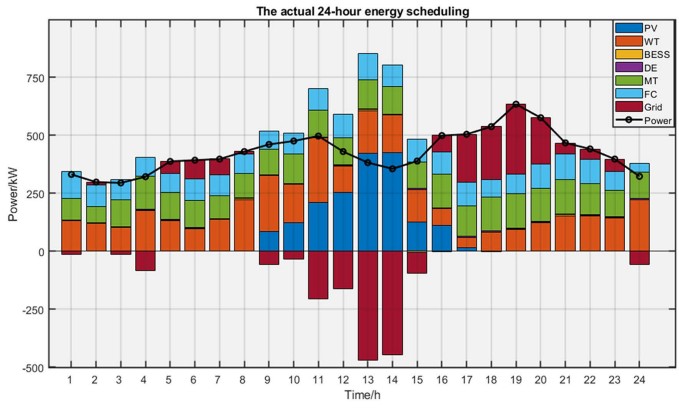
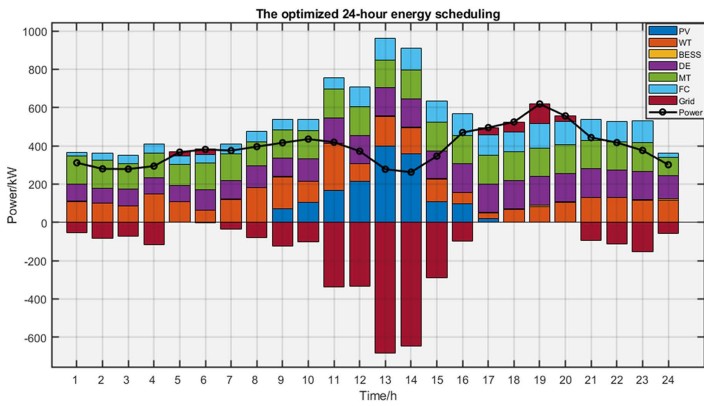

(a) **Actual scheduling**                    (b) **Optimized scheduling**

**Fig 8. Comparison of energy scheduling.**

hikes. As the growth rate escalates, overall emissions plateau due to production limitations, while carbon trading expenses persist in their ascent.

The reward-penalty coefficient directly modifies the framework of tiered cost computation. For coefficients ranging from 0 to 0.1, minimal output modifications comply with carbon allowances while exhibiting incremental cost increases. At 0.2–0.5, multi-tier expenses require substantial scheduling modifications, resulting in fast escalations in trade costs. Above 0.5, high-tier prices and equipment limits prevent emission optimization, stabilizing emissions despite rising costs, demonstrating an equilibrium between equipment constraints and carbon penalty mechanisms. At this juncture, it solely serves to elevate carbon transaction costs, which can no longer fulfill the emission reduction objectives. It illustrates the equilibrium of the game between equipment scheduling constraints and carbon trading penalties.

### 5.4 Daily energy scheduling analysis

This section analyzes the daily energy scheduling structure and compares the performance of the two mechanisms. The results of the traditional scheduling strategy and the improved scheduling strategy are shown in Fig 8.

Figure (a) and (b) reveal that conventional scheduling prioritizes system stability by suppressing renewable generation to minimize energy mix uncertainty. However, solar and wind resources possess significant untapped potential. The optimized strategy maximizes renewable utilization, concentrating solar generation during peak insolation (11:00–14:00) and wind during stable periods (20:00–24:00). This reduces fossil fuel dependence, lowering both carbon emissions and operational costs. Crucially, the strategy caps renewable utilization at 88%, maintaining flexibility to mitigate variability risks.

Traditional scheduling over-relies on diesel/gas turbines, frequently running them at full capacity. The optimized strategy balances regenerables and fossil fuels, using the latter only when renewables fall short. Conventional energy storage has erratic charge/discharge patterns and low efficiency, failing to stabilize demand. The improved strategy coordinates storage to charge in low-demand periods and discharge during peaks, effectively reducing demand swings. By increasing regenerables and cutting emissions, the system qualifies for carbon trading and green certificate schemes, turning environmental benefits into revenue and lowering overall costs.

Based on the above analysis, this paper proposes scheduling recommendations aimed at prioritizing renewable energy utilization, restricting fossil fuel use, and dynamically deploying energy storage:

1) Renewable Priority: During high solar/wind availability, dispatch renewables at maximum capacity using predictive models (e.g., weather forecasts). Limit diesel/gas turbine activation.

2) Storage Optimization: Charge storage when renewable output exceeds 80% of real-time demand or when grid prices fall below 0.3 CNY/kWh. Discharge when the net demand deficit exceeds 10% or grid prices surpass 0.8 CNY/kWh. Set charge/discharge thresholds to prevent inefficiency.

3) Fossil Fuel Restrictions: Limit diesel/gas turbines to ≤4 hours daily operation at 30–50% rated power. Activate only when storage state-of-charge (SOC) drops below 20%. Dynamically switch between storage and turbines based on emissions monitoring (e.g., $SO_2/NO_x$).

4) Economic Integration: Inject surplus renewables (>120% of demand) into the grid during peak generation (e.g., mid-day). Utilize price signals to buy electricity at low-cost periods (<0.3 CNY/kWh) and sell during shortages/high prices (>0.8 CNY/kWh), transforming the grid into a revenue stream while reducing emissions.

## 6 Conclusion

This research presents an innovative low-carbon scheduling paradigm for integrated energy systems. An upgraded green certificate carbon trading mechanism, dynamic reward and punishment elements, and a multi-strategy goose optimization method are used. The principal contributions are:

1] The dynamic GCT-CET mechanism addresses China's carbon market design barrier by fulfilling the 'high renewable consumption—deep decarbonization—toxic emission reduction' trilemma in IES through a policy-compatible method. Linking reward-penalty elements to real-time renewable ratios (Section 2.2.1) reduces carbon by 27.3% without sacrificing economic efficiency (Section 5.2), giving policymakers a scalable dynamic carbon pricing approach.

2] The MSAGOOSE, enhanced by multimodal distribution-guided search and group-aware opposition-based learning, surpasses seven advanced approaches by one to two orders of magnitude in accuracy. It guarantees strong convergence in resolving high-uniqueness issues and effectively tackles the 168-dimensional IES scheduling problem in 155 seconds, fulfilling practical engineering requirements.

3] Analysis of coefficient changes shows that carbon trading parameters greatly impact IES carbon emission reduction and cost-effectiveness, promoting low-carbon economic dispatch.

4] Comparisons between traditional and improved scheduling strategies yield practical suggestions for real-world energy scheduling. For instance, during solar peak hours (11:00–14:00), when renewable surplus exceeds 80% or the electricity price is below 0.3 CNY/kWh, deploying 30–50% energy storage and limiting fossil generators to ≤4h/day is recommended.

Future work will quantify carbon price fluctuation impacts on scheduling stability, explore multi-time-scale optimization, and extend this framework to multi-regional carbon markets and electric vehicle-grid integration. Overall, this research supports IES planning and operation under the "double carbon" target, proving the effectiveness of combining market mechanisms and intelligent algorithms.

## Supporting information

**S1 Fig. All the pictures in the article.**
(ZIP)

**S1 Table. All the tables in the article.**
(DOCX)

**S1 Data. Original data.**
(XLSX)

## Author contributions

**Conceptualization:** Chengcheng Ding, zu Yun.

**Data curation:** Chengcheng Ding.

**Formal analysis:** Chengcheng Ding.

**Funding acquisition:** Chengcheng Ding, Zzu Yun.

**Investigation:** Chengcheng Ding.

**Methodology:** Chengcheng Ding.

**Project administration:** Chengcheng Ding.

**Resources:** Chengcheng Ding.

**Software:** Chengcheng Ding.

**Supervision:** Chengcheng Ding, Zhu Yun.

**Validation:** Chengcheng Ding, Zhu Yun.

**Visualization:** Chengcheng Ding.

**Writing – original draft:** Chengcheng Ding.

**Writing – review & editing:** Chengcheng Ding, Zhu Yun.

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
