## [Decision Letter · Decision Letter 0]

22 Jul 2025

Dear Dr. YUN,

Thank you for submitting your manuscript to PLOS ONE. After careful consideration, we feel that it has merit but does not fully meet PLOS ONE’s publication criteria as it currently stands. Therefore, we invite you to submit a revised version of the manuscript that addresses the points raised during the review process.

**ACADEMIC EDITOR:** please revise accordingly

We look forward to receiving your revised manuscript.

Kind regards,

Zhengmao Li

Academic Editor

PLOS ONE

Journal Requirements:

**Additional Editor Comments:**

please revise accordingly

Reviewers' comments:

Reviewer's Responses to Questions

**Comments to the Author**

1. Is the manuscript technically sound, and do the data support the conclusions?

Reviewer #1: Yes

Reviewer #2: Yes

2. Has the statistical analysis been performed appropriately and rigorously?

Reviewer #1: Yes

Reviewer #2: Yes

3. Have the authors made all data underlying the findings in their manuscript fully available?

Reviewer #1: Yes

Reviewer #2: Yes

4. Is the manuscript presented in an intelligible fashion and written in standard English?

Reviewer #1: Yes

Reviewer #2: Yes

Reviewer #1: The manuscript presents a timely and relevant study on low-carbon optimal scheduling of integrated energy systems by incorporating a dynamic green certificate–carbon trading mechanism and an improved swarm intelligence algorithm. The topic aligns well with current global efforts toward achieving carbon neutrality and provides a meaningful contribution to the area of low-carbon dispatch strategies. However, to enhance the clarity, scientific rigor, and generalizability of the work, several aspects need further attention. :

1. The abstract is overly lengthy and, although information-dense, lacks clear logical organization. It is recommended to adopt a four-part structure — Background, Method, Results, and Conclusion — to present the content more clearly.

2. It is recommended that the authors enrich the Introduction by incorporating recent studies on multi-agent game-theoretic approaches—such as asymmetric Nash bargaining, and Cournot–Nash competition—in the context of integrated energy systems. This would broaden the modeling perspective and provide a solid theoretical foundation for future research directions..

[1] A Distributed Market-Aided Restoration Approach of Multi-Energy Distribution Systems Considering Comprehensive Uncertainties from Typhoon Disaster.

[2] Risk-averse stochastic capacity planning and P2P trading collaborative optimization for multi-energy microgrids considering carbon emission limitations: An asymmetric Nash bargaining approach.

3. Although the optimization model includes power balance and ramping constraints, it omits essential practical constraints such as SOC limits for storage and dynamic bounds for generation. Please enrich the model description to enhance engineering fidelity.

4. The manuscript lacks a unified modeling framework diagram. It is recommended to include a schematic diagram that connects the incentive mechanism, scheduling model, and optimization algorithm to highlight the closed-loop structure of policy response and optimization.

5. Algorithm performance testing should clarify how hyperparameters were tuned, how convergence was judged, and whether statistical significance was verified across runs.

Reviewer #2: 1.Although the introduction cites many references, few of them directly support the research motivation or identified limitations. It is recommended to streamline general references and supplement targeted literature on scheduling mechanisms, carbon market integration, and algorithmic performance issues.

2.Section 2.2 lacks detailed analysis and structural elaboration of the proposed GCT-CET interaction mechanism. The mechanism diagram is directly adapted from prior literature, and the described improvements are vague.

3.Section 3.2 suffers from fragmented organization. The multiple strategy modules lack a coherent design logic and are presented in isolation without clarifying their overall purpose, applicable stages, or interrelations.

4.Section 4.1 introduces CEC benchmark function testing to evaluate algorithm performance; however, the logical connection between this part and the integrated energy system scheduling problem is unclear. The testing objective appears disconnected from the main model, making this section seem isolated within the overall structure.

**Do you want your identity to be public for this peer review?** For information about this choice, including consent withdrawal, please see our Privacy Policy

Reviewer #1: No

Reviewer #2: No

---

## [Author Response · Author response to Decision Letter 1]

12 Aug 2025

Manuscript Low-carbon optimal scheduling of integrated energy systems based on multi-strategy ameliorated goose algorithm and green certificate-carbon trading coordination

Manuscript ID�PONE-D-25-37059

Dear Editor and Reviewers,

We gratefully appreciate the editor and all reviewers for their time spent making positive and constructive comments. These comments are all valuable and helpful for revising and improving our manuscript entitled “Low-carbon optimal scheduling of integrated energy systems based on multi-strategy ameliorated goose algorithm and green certificate-carbon trading coordination” (ID: PONE-D-25-37059) for publication in PLOS ONE, as well as the important guiding significance to our research. We have carefully considered the feedback from the reviewers and have made substantial revisions to enhance the clarity, coherence, and scientific rigor of our work.

The revised sections are highlighted in yellow. Attached please find the revised version, which we would like to submit for your kind consideration.

Thank you once again for the opportunity to improve our manuscript. We look forward to the possibility of our revised work being published in PLOS ONE.

Best regards.

Sincerely,

Chengcheng Ding

Yun ZHU

The following is a point-to-point response to reviewers’ comments.

Response to reviewers:

Reviewer #1:

1. The abstract is overly lengthy and, although information-dense, lacks clear logical organization. It is recommended to adopt a four-part structure — Background, Method, Results, and Conclusion — to present the content more clearly.

Authors’ Answer: Thank you very much to the expert for your valuable comments.

The author has thoroughly rewritten the abstract using the recommended four-part structure. The revised abstract now concisely covers the background, methodology, main results, and conclusions, thereby improving readability and logical flow. The author has rewritten the abstract as follows (Page 1, Line 10 to Line 26):

Background:

Integrated energy systems (IES) in China face a dual challenge under the “dual-carbon” targets: maximizing renewable energy utilization while minimizing carbon emissions. Traditional tiered carbon markets often lack the flexibility to dynamically incentivize low-carbon operation.

Method:

To address this, a coordinated framework is proposed, integrating a dynamic carbon emission trading (CET) mechanism with green certificate trading (GCT) and a Multi-Strategy Ameliorated Goose Optimization (MSAGOOSE) algorithm. The GCT-CET mechanism introduces exponential reward–penalty coefficients based on real-time renewable consumption rates, enabling adaptive carbon pricing. MSAGOOSE combines adaptive parameter adjustment, multimodal distribution-guided exploration, and population-aware reverse learning to improve optimization robustness in high-dimensional, nonlinear scheduling problems.

Results:

Benchmark evaluations on CEC2017 and CEC2022 show that MSAGOOSE achieves an order-of-magnitude improvement in accuracy over seven state-of-the-art algorithms. In a 24-hour IES scheduling case in Anhui Province, the proposed method reduces carbon emissions by 27.3% (5,121 kg/d), increases renewable energy share to 88%, and cuts operating costs by 24.8% (6,151 CNY/d). Parametric analysis further confirms the framework’s effectiveness in balancing economic and environmental goals under decentralized energy scenarios.

Conclusion:

This study presents a policy-algorithm co-design paradigm that offers both theoretical and practical support for low-carbon IES transitions, enabling scalable, flexible, and economically viable scheduling strategies.

2. It is recommended that the authors enrich the Introduction by incorporating recent studies on multi-agent game-theoretic approaches—such as asymmetric Nash bargaining, and Cournot–Nash competition—in the context of integrated energy systems. This would broaden the modeling perspective and provide a solid theoretical foundation for future research directions.

[1] A Distributed Market-Aided Restoration Approach of Multi-Energy Distribution Systems Considering Comprehensive Uncertainties from Typhoon Disaster.

[2] Risk-averse stochastic capacity planning and P2P trading collaborative optimization for multi-energy microgrids considering carbon emission limitations: An asymmetric Nash bargaining approach.

Authors’ Answer: Thank you very much to the expert for your valuable comments.The literatures proposed by the experts are very forward-looking and creative. The author quotes these documents and highly praises the contribution of these documents to the development of the current energy industry. The details are as follows:

(1) Reference [6,7], based on a non-Nash equilibrium, incorporates electricity-heat bidding within a distributed market-assisted recovery framework for multi-energy systems, providing a creative template for IES scheduling. (Page 2, line 43 to line 45)

[6] Wang Z, Hou H, Wei R, et al. A Distributed Market-Aided Restoration Approach of Multi-Energy Distribution Systems Considering Comprehensive Uncertainties from Typhoon Disaster[J]. IEEE Transactions on Smart Grid, 2025.

[7] Hou H, Wang Z, Zhao B, et al. Peer-to-peer energy trading among multiple microgrids considering risks over uncertainty and distribution network reconfiguration: A fully distributed optimization method[J]. International Journal of Electrical Power & Energy Systems, 2023, 153: 109316.

(2) The research suggests using asymmetric Nash bargaining to distribute benefits fairly in carbon market transactions, and its "contribution-bargaining ability" mapping technique may help integrate dynamic carbon markets (Page 2, line 63 to line 66)

[26] Wang Z, Hou H, Zhao B, et al. Risk-averse stochastic capacity planning and P2P trading collaborative optimization for multi-energy microgrids considering carbon emission limitations: An asymmetric Nash bargaining approach[J]. Applied Energy, 2024, 357: 122505.

3. Although the optimization model includes power balance and ramping constraints, it omits essential practical constraints such as SOC limits for storage and dynamic bounds for generation. Please enrich the model description to enhance engineering fidelity.

Authors’ Answer: Thank you very much to the expert for your valuable comments. In order to enhance the fidelity and reproducibility of the research, the author enriches the description of relevant constraints and supplements the constraint function model in section 2.2. In section 5, the related parameters, such as the stored SOC limit and the dynamic limit of power generation, are supplemented. Details are as follows:

1) Model supplement (Page 7, line 215 to line 221):

(2) Power output constraint

In the formula, and represent the upper and lower limits of each power output. represents the limits of each power output.

(3) Energy storage operation constraint

In the formula, and are the minimum and maximum output power of the energy storage unit, respectively; and are the lower and upper limits of the capacity of the energy storage unit, respectively.

2) Data imputation (Page 16-17, line 399 to line 401)

Table 7. Related operating parameters of power supply equipment

Type DE MT FC Grid

Lower power limit/kW 3 3 3 -300

Upper power limit/kW 150 150 150 300

Limit of climbing power/(kW/min) 1.5 1.5 1.5

Operation coefficient (CNY/kW) 0.128 0.0489 0.0288

Table 8. Energy storage parameters

Type Parameter Value Parameter Value

BESS Socmax/ (kW·h) 150 PBESS, min / kW -150

Socmin / (kW·h) 0 Efficiency 0.9

PBESS, max / kW 150

4. The manuscript lacks a unified modeling framework diagram. It is recommended to include a schematic diagram that connects the incentive mechanism, scheduling model, and optimization algorithm to highlight the closed-loop structure of policy response and optimization.

Authors’ Answer: Thank you very much for your valuable and constructive comments. The author combs through the whole article in detail, and expands the flow chart of the original algorithm, connects the schematic diagram of the incentive mechanism, scheduling model, and optimization algorithm, and constructs a closed-loop structure that highlights policy response and optimization. The final results are as follows (Page 11, line 320):

Fig 3. The flow chart of the IES program solved by the MSAGOOSE algorithm

5. Algorithm performance testing should clarify how hyperparameters were tuned, how convergence was judged, and whether statistical significance was verified across runs.

Authors’ Answer: Thank you very much for your valuable and constructive comments. For hyperparameter adjustment, the author joined the relevant introduction. For the problem of convergence judgment, the author supplements the formula of convergence judgment and its introduction. In order to verify the statistical significance, the author added the p and the Wilcoxon rank-sum test experimental data into Table 4, and verified the superiority and reliability of the improved algorithm to ensure the reproducibility of the results. The statistical significance was verified. The specific results are as follows

1)Hyperparameter description

The selection of hyperparameters is guided by the literature and CEC test function, and screened by the grid experiment method to ensure its reliability and portability in different scenarios. (Page 12, line 342 to line 344)

Table 2 Hyperparameters of various algorithms (Page 12-13, line 348)

Algorithm Hyperparameters Convergence Criteria

GA p_c = 0.9, p_m = 0.01 Loops = Maxit or or δ< 1e-6 over 30 iterations

TDBO P= 0.2; k = 5; w1 = 0.5; w2 =0.1

Loops = Maxit or or δ< 1e-6 over 30 iterations

GSWOA w_max∈[0.2, 0.5]; b_base=0.75; P=∈[0.3, 0.7] Loops = Maxit or or δ< 1e-6 over 30 iterations

LEA h_max = 0.7; h_min = 0; λ_c =0.5; λ_p=0.5 Loops = Maxit or or δ< 1e-6 over 30 iterations

CPO T = 2�step = 0.02�α = 0.2�Tf = 0.8 Loops = Maxit or or δ< 1e-6 over 30 iterations

HO b∈[2, 4]; c∈[1, 1.5]; d∈[2, 3]; l∈[-2π, 2π] Loops = Maxit or or δ< 1e-6 over 30 iterations

GOOSE coe=0.83; Weight_Stone=12; pro=0.2; rnd=0.5 Loops = Maxit or or δ< 1e-6 over 30 iterations

MSAGOOSE coe=0.83; Weight_Stone=12; pro=0.2;rnd=p Loops = Maxit or or δ< 1e-6 over 30 iterations

2)Judge convergence (Page 8, line 249 to line 253):

The convergence condition of the algorithm is

When the relative change rate of the target value converges or the number of cycles is greater than the cycle ends and is judged to be convergent. This is one of the most commonly used criteria for iterative algorithms.

3)Verify statistical significance (Page 14, line 354 to line 357):

Table 4. Experimental results on CEC2022 and CEC2017 (Page 13-14, line 352)

Functons Agorithm

GA S2 TDBO S1 GSWOA HO CPO LEA GOOSE MSAGOOSE

F4 Opt 898 841 899 856 907 839 916 855 891 800

Ave 933 858 923 882 931 886 921 873 901 801

Std 1.43 15.27 17.08 13.9 21.86 10.22 26.18 15.17 23.1 24.42

Fri 8.7 2.75 7.9 4.15 8.8 4.35 7.9 3.45 5.95 1.05

Rank 9 2 7 4 10 5 8 3 6 1

p 2.33E-11 2.33E-11 2.33E-11 2.33E-11 2.33E-11 2.33E-11 2.33E-11 2.33E-11 2.33E-11

F8

Opt 2256 2223 2246 2227 2229 2239 2240 2249 2455 2200

Ave 2299 2259 2251 2229 2238 2241 2241 2278 2531 2201

Std 63.82 57.11 6.11 5.28 7.84 9.43 2.93 53.95 70.52 0.33

Fri 7.05 4.10 7.30 2.85 5.00 5.20 5.95 6.55 10.00 1.00

Rank 8 3 9 2 4 5 6 7 10 1

p 2.96E-07 2.96E-07 2.96E-07 2.96E-07 2.96E-07 2.96E-07 2.96E-07 2.96E-07 2.96E-07

F12 Opt 2914 2801 2795 2836 2947 2837 3105 2892 4601 2700

Ave 3158 2890 2884 2889 3533 2899 2962 2861 5468 2732

Std 72.12 113.36 32.57 44.20 220.85 42.64 12.12 21.01 387.59 9.65

Fri 8.00 3.50 4.20 4.30 9.00 4.60 6.55 3.15 10.00 1.70

Rank 8 3 4 5 9 6 7 2 10 1

p 3.02E-11 3.82E-09 3.02E-11 3.82E-09 3.02E-11 3.02E-11 3.02E-11 3.02E-11 3.02E-11

F7 Opt 2250 739 2460 1486 3012 2579 1778 2317 2392 703

Ave 2436 799 2533 1613 3055 2632 1812 2431 2449 715

Std 370.73 80.28 146.92 253.24 86.19 105.38 67.88 227.07 112.70 76.23

Fri 6.57 1.87 7.10 3.23 9.90 8.13 3.77 6.73 6.57 1.17

Rank 5 2 8 3 10 9 4 7 6 1

p 3.02E-11 3.02E-11 3.02E-11 3.02E-11 3.02E-11 3.02E-11 3.02E-11 3.02E-11 3.02E-11

F19 Opt 2049 2009 2001 1979 1988 1991 1991 2028 2281 1951

Ave 8.08E+06 3.85E+03 9.08E+08 1.82E+06 2.08E+10 1.63E+07 3.56E+07 1.28E+07 8.04E+05 2.00E+03

Std 1.23E+06 2.43E+03 2.45E+07 1.69E+05 2.08E+08 9.94E+05 1.14E+06 6.01E+05 1.08E+05 1.04E+03

Fri 5.33 1.83 9.00 3.93 10.00 6.43 7.77 6.27 3.27 1.17

Rank 5 2 9 4 10 7 8 6 3 1

p 3.02E-11 1.86E-10 3.02E-11 1.86E-10 7.69E-11 3.02E-11 3.02E-11 3.02E-11 3.02E-11

F30 Opt 6.38E+06 5.09E+03 8.60E+08 7.23E+05 1.69E+10 1.24E+08 1.96E+08 1.84E+07 2.02E+06 5.09E+03

Ave 4.90E+07 1.01E+04 2.32E+09 2.09E+07 2.36E+10 2.69E+08 3.35E+08 7.23E+07 2.91E+06 9.98E+03

Std 3.08E+07 5.86E+03 7.53E+08 2.35E+07 5.26E+09 9.75E+07 8.60E+07 2.96E+07 4.64E+05 5.43E+03

Fri 5.10 1.90 9.00 4.13 10.00 7.23 7.77 5.63 3.13 1.10

Rank 5 2 9 4 10 7 8 6 3 1

p 1.88E-11 5.83E-10 2.56E-11 5.83E-10 1.88E-11 2.41E-11 2.51E-10 1.66E-11 1.88E-10

The p denotes the Wilcoxon rank-sum test, a non-parametric check; when p < 0.05, we reject the null hypothesis and deem the improvement statistically significant rather than random. All p-values are far below 1 × 10⁻⁶, confirming MSAGOOSE’s superiority is highly significant across repeated runs.

Reviewer #2:

1. Although the introduction cites many references, few of them directly support the research motivation or identified limitations. It is recommended to streamline general references and supplement targeted literature on scheduling mechanisms, carbon market integration, and algorithmic performance issues.

Authors’ Answer: Thank you very much for your valuable and constructive comments. In response to this problem, the author re-combed the content of the introduction, and re-analyzed the references cited in this paragraph according to the core of each paragraph, and deleted the original seven documents with weak support and honor. Among them, the references [1],[4], and [5] mainly provide macro policy and technical trend background, but it is not directly related to the core GCT-CET mechanism construction, IES scheduling modeling, and algorithm improvement. Although it is helpful to introduce the background, too many references in the introduction will scatter the pertinence of the research motivation, so it is streamlined. references [ 6 ] are also redundancy problems; in order to simplify the literature, delete. References [20], [21] and [24] mainly introduce the algorithm, which are not related to the IES in this paper, and are also deleted.

At the same time, in order to strengthen the support of research motivation, on the basis of streamlining the literature, the author re-read and collected targeted literature on scheduling mechanisms, carbon market integration and algorithm performance issues. After careful screening, some relatively good literatures were selected to support this study.Then, we include (1) references [6], [7], [9 ], [10], [12] and [13] on the scheduling mechanism of integrated energy system ( IES ), including the existing mechanism innovation and the problem of its scheduling mechanism, which directly connects the research direction of the scheduling mechanism in this paper ; ( 2 ) references [22], [23], [25] and [26] directly link the carbon market mechanism with scheduling, including dynamic GCT-CET interaction and market optimization under emission constraints, pointing out the rigidness of the current two mechanisms and the narrowness of the linkage mechanism ; ( 3 ) The latest progress in the performance of swarm intelligence algorithm, GOOSE original formula and multi-strategy improvement of high-dimensional optimization in References [27] - [32]. These additions directly strengthen the theoretical and methodological foundations of our proposed models and algorithms, ensuring that the cited work now provides accurate support for research motivation, mechanism design, and performance verification.

The specific results are as follows

1)Redundant literature deletion

[1] J. Peng, J. Wang, J. Qi, and Y. Liao

---

## [Decision Letter · Decision Letter 1]

24 Aug 2025

Low-carbon optimal scheduling of integrated energy systems based on multi-strategy ameliorated goose algorithm and green certificate-carbon trading coordination

PONE-D-25-37059R1

Dear Dr. YUN,

We’re pleased to inform you that your manuscript has been judged scientifically suitable for publication and will be formally accepted for publication once it meets all outstanding technical requirements.

Kind regards,

Zhengmao Li

Academic Editor

PLOS ONE

Additional Editor Comments (optional):

Reviewers' comments:

Reviewer's Responses to Questions

**Comments to the Author**

Reviewer #1: All comments have been addressed

Reviewer #2: All comments have been addressed

2. Is the manuscript technically sound, and do the data support the conclusions?

Reviewer #1: Yes

Reviewer #2: Yes

3. Has the statistical analysis been performed appropriately and rigorously?

Reviewer #1: Yes

Reviewer #2: Yes

4. Have the authors made all data underlying the findings in their manuscript fully available?

Reviewer #1: Yes

Reviewer #2: Yes

5. Is the manuscript presented in an intelligible fashion and written in standard English?

Reviewer #1: Yes

Reviewer #2: Yes

Reviewer #1: (No Response)

Reviewer #2: The authors have addressed all previous concerns satisfactorily, and the revised manuscript is acceptable.

**Do you want your identity to be public for this peer review?** For information about this choice, including consent withdrawal, please see our Privacy Policy

Reviewer #1: No

Reviewer #2: No

---

## [Editor Report · Acceptance letter]

PONE-D-25-37059R1

PLOS ONE

Dear Dr. YUN,

I'm pleased to inform you that your manuscript has been deemed suitable for publication in PLOS ONE. Congratulations! Your manuscript is now being handed over to our production team.

Kind regards,

on behalf of

Dr Zhengmao Li

Academic Editor

PLOS ONE